# Vaccine effectiveness of CoronaVac against COVID-19 among children in Brazil during the Omicron period

Pilar T. V. Florentino [1,2] ✉, Flávia J. O. Alves [1], Thiago Cerqueira-Silva [3,4], Vinicius de Araújo Oliveira [1,4], Juracy B. S. Júnior[5], Adelson G. Jantsch[3], Gerson O. Penna [6], Viviane Boaventura[3,4], Guilherme L. Werneck [7,8], Laura C. Rodrigues[9], Neil Pearce[9], Manoel Barral-Netto [1,4], Mauricio L. Barreto[1,10] & Enny S. Paixão [9,10]

Although severe COVID-19 in children is rare, they may develop multisystem inflammatory syndrome, long-COVID and downstream effects of COVID-19, including social isolation and disruption of education. Data on the effectiveness of the CoronaVac vaccine is scarce during the Omicron period. In Brazil, children between 6 to 11 years are eligible to receive the CoronaVac vaccine. We conducted a test-negative design to estimate vaccine effectiveness using 197,958 tests from January 21, 2022, to April 15, 2022, during the Omicron dominant period in Brazil among children aged 6 to 11 years. The estimated vaccine effectiveness for symptomatic infection was 39.8% (95% CI 33.7–45.4) at ≥14 days post-second dose. For hospital admission vaccine effectiveness was 59.2% (95% CI 11.3–84.5) at ≥14 days. Two doses of CoronaVac in children during the Omicron period showed low levels of protection against symptomatic infection, and modest levels against severe illness.

Randomized clinical trials have demonstrated high mRNA vaccine efficacy and immunogenicity in children and adolescents[1, 2]. However, data related to the inactivated-virus vaccine (CoronaVac) of efficacy and effectiveness (VE) against the SARS-CoV-2 B.1.1.529 (Omicron) variant are lacking for children aged 6–11 years.

Although severe COVID-19 is a rare condition in children[3], the widespread distribution of SARS-CoV-2 infection and the increasing number of cases in this population has caused a significant public health impact. Besides, children are also susceptible to the multisystem inflammatory syndrome in Children (MIS-C), long-COVID syndrome[3, 4] and downstream effects of COVID-19, including social isolation and interruption in education[4]. Therefore, there is an urgent need to collect more data on the effectiveness of vaccines, especially in the Omicron period, to guide decision-makers in adopting policies, such as mandating mask use in school settings.

In Brazil, the children's vaccination campaign started on January 21, 2022[5], and CoronaVac has been used for children aged 6–11 years. On April 15, 2022, vaccine uptake for all vaccines used in children was 62.9% for the 1st dose and 26.6% for the second dose. For CoronaVac, vaccine uptake was 35.1% for 1st dose and 19.8% for the second dose. To our knowledge, no report estimates vaccine effectiveness for CoronaVac among children aged 6–11 years during the Omicron period. Therefore, in this observational study using a nationwide database from Brazil, we estimated the vaccine effectiveness (VE) of the

[1]Centre of Data and Knowledge Integration for Health (CIDACS), Instituto Gonçalo Moniz Institute, Oswaldo Cruz Foundation (Fiocruz), Salvador, Brazil. [2]Biomedical Science Institute, University of São Paulo, São Paulo, Brazil. [3]Gonçalo Moniz Institute, Oswaldo Cruz Foundation (Fiocruz), Salvador, Brazil. [4]Faculty of Medicine, Federal University of Bahia, Salvador, Brazil. [5]Public Health Institute, Federal University of Bahia, Salvador, Brazil. [6]Tropical Medicine Centre, University of Brasília, Fiocruz School of Government, Brasília, Brazil. [7]Department of Epidemiology, Social Medicine Institute, State University of Rio de Janeiro, Rio de Janeiro, Brazil. [8]Institute of Collective Health Studies, Federal University of Rio de Janeiro, Rio de Janeiro, Brazil. [9]London School of Hygiene and Tropical Medicine, London, UK. [10]These authors jointly supervised this work: Mauricio L. Barreto, Enny S. Paixão. ✉e-mail: pilar.veras@gmail.com

CoronaVac against medically attended symptomatic and severe COVID-19 in children aged 6–11 years.

## Results

During the study period, 197,958 tests were performed on Brazilian children aged 6–11 years, with 89,595 (45.3%) cases and 108,363 (54.7%) controls, with 508 hospital admissions (Fig. S1). The age, sex, geographic region, socioeconomic position, comorbidities, and hospital admission were similar among the children who tested positive and negative (Table S1). For children between 6 and 11 years, VE against symptomatic COVID-19 during Omicron circulation was 21.2% (95% CI 18.6–23.8) after 13 days post first dose of CoronaVac. After the second dose, VE reached 30.8% (95% CI 24.2–36.8) at 0–13 days and 39.8% (95% CI 33.7–45.4) at ≥14 days (Table 1; Fig. 1) with most of the individuals being tested within 43 days after the second dose (Figure S2). For hospital admission among children vaccinated with one dose of CoronaVac at ≥14 days, the adjusted VE was 47.1% (95% CI 26.6–62.7). After two doses of CoronaVac, the adjusted VE was 82.4% (95% CI 44.2–97.1) at 0–13 days and 59.2% (95% CI 11.3–84.5) at ≥14 days (Table 1; Fig. 1). For ICU admission there were two cases among children vaccinated with two-dose at ≥14 days and the estimated VE for rare events was 20.9% (95% CI [−177.2]–85.0) (Table S2). No death events were detected among children vaccinated with two doses. The sensitivity analyses using multiple imputations for missing data in ethinicity (19.4%) produced similar results to the primary analyses (Table S3). Furthermore, the analyses excluding the previously infected group generated similar VE estimates (Table S4).

## Discussion

In this investigation of CoronaVac VE in children 6–11 years of age during Omicron variant predominance, we found that two doses of the CoronaVac vaccine were 39.8% effective against medically attended symptomatic COVID-19 and 59.2% effective in preventing hospital admission COVID-19 cases at ≥14 days after the second dose. The VE estimated in children 6–11 years in Brazil during the Omicron period was much lower than the effectiveness of 75.8% reported for the same demographic in Chile when B.1.617.2 (Delta) was the predominant circulating SARS-CoV-2 variant[6]. However, our data were comparable with results observed in children aged 3–5 during the Omicron outbreak in the same country, 38.2%; (95% CI, 36.5–39.9) against symptomatic disease and, 64.6% (95% CI, 49.6–75.2) against hospitalisation[7]. These findings are also in line with previous studies of VE in adult and

adolescent populations that have shown a significant reduction in VE against Omicron compared with early pandemic variants[8, 9]. Although we have analysed VE at the optimal period of the second dose among children vaccinated with CoronaVac, it is likely to wane quickly, especially during the Omicron period as it was seen for the adolescent and children population vaccinated with BNT162b2[8, 10–13].

This study has strengths and limitations. A strength of this study is the high-quality nationwide database from Brazil. Furthermore, we used Test Negative Design (TND) to minimise bias related to access to health care and health-seeking behaviour. TND's primary assumption is that people seeking and getting tested would be influenced by similar pressures regardless of vaccination status[14]. Another strength is the improbable under ascertainment of vaccination status since the all-vaccines doses administered against COVID-19 in Brazil are recorded in the national immunisation system (SI-PNI). An important limitation is the high rates of asymptomatic infection allied to limited testing in Brazil among children since the database from the study only accounts for tests from the healthcare system and not community testing. Also, the under ascertainment of previous infection may bias the VE estimates if this condition occurs differentially or non-differentially in the vaccinated and unvaccinated group[15–17].

In summary, our findings indicate low levels of protection against symptomatic infection with the Omicron variant after two doses of vaccination with CoronaVac among children. Hence, in line with previous studies involving other vaccines and age groups, the vaccination program alone is unlikely to suppress viral circulation. However, this vaccine was 59.2% effective against COVID-19-hospital admissions, albeit with wide uncertainty intervals. Further studies will be necessary to assess the duration of protection, specially against complications of COVID-19 that occur in the pediatric population, such as MIS-C and long-COVID. Effectiveness also must continue to be monitored as new variants arise.

## Methods
### Data sources

Data were obtained from three routinely collected sources: the national surveillance system for RT-PCR and antigen tests for COVID-19 infection (e-SUS Notifica); the information system for severe acute respiratory illness (SIVEP-Gripe). These two datasets present notifications from public and private healthcare systems of SARS-CoV-2 suspected cases, and hospitalisation cases of SARS, respectively. Also, the national immunisation system (SI-PNI).

**Table 1 | Odds Ratio and Vaccine Effectiveness for Symptomatic Infection and Hospital admission among children aged 6–11 vaccinated with Coronavac**

| Symptomatic infection | | | | | |
|---|---|---|---|---|---|
| Vaccination status | Positive tests $n = 89,595$ | Negative tests $n = 108,363$ | OR Crude (95% CI) | OR adjusted (95% CI) | VE (%) (95% CI) |
| Unvaccinated | 72,737 (50.99%) | 69,923 (49.01%) | | | |
| 1st dose | | | | | |
| 0–13 days | 7499 (52.22%) | 6862 (47.78%) | 1.05 (1.02, 1.09) | 1.09 (1.05, 1.13) | [−9.0 (−13.1, −4.9)] |
| ≥14-2nd dose | 8205 (28.89%) | 20,193 (71.11%) | 0.39 (0.38, 0.40) | 0.79 (0.76, 0.81) | 21.2 (18.6, 23.8) |
| 2nd dose | | | | | |
| 0–13 days | 630 (12.16%) | 4552 (87.84%) | 0.13 (0.12, 0.14) | 0.69 (0.63, 0.76) | 30.8 (24.2, 36.8) |
| ≥14 days | 524 (7.12%) | 6833 (92.88%) | 0.07 (0.07, 0.08) | 0.60 (0.55, 0.66) | 39.8 (33.7, 45.4) |
| Hospital admission | | | | | |
| Vaccination status | Positive tests $n = 508$ | Negative tests $n = 108,363$ | OR Crude (95% CI) | OR adjusted (95% CI) | VE (%) (95% CI) |
| Unvaccinated | 428 (0.61%) | 69,923 (99.39%) | | | |
| 1st dose | | | | | |
| 0–13 days | 30 (0.44%) | 6862 (99.56%) | 0.71 (0.49, 1.04) | 0.73 (0.49, 1.05) | 27.0 (−5.2, 51.1) |
| ≥14-2nd dose | 42 (0.21%) | 20193 (99.79%) | 0.34 (0.25, 0.47) | 0.53 (0.37, 0.73) | 47.1 (26.6, 62.7) |
| 2nd dose | | | | | |
| 0–13 days | 2 (0.04%) | 4552 (99.96%) | 0.07 (0.02, 0.29) | 0.18 (0.03, 0.56) | 82.4 (44.2, 97.1) |
| ≥14 days | 6 (0.09%) | 6833 (99.91%) | 0.14 (0.06, 0.32) | 0.41 (0.16, 0.89) | 59.2 (11.3, 84.5) |

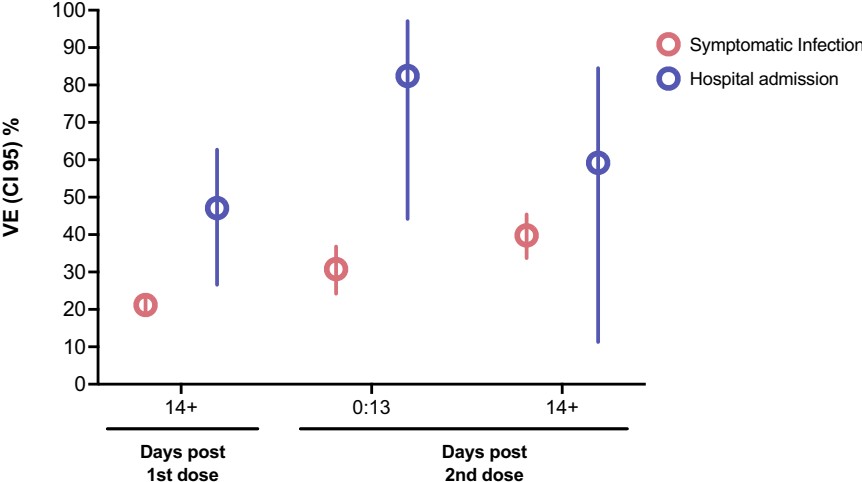

**Fig. 1 | Vaccine Effectiveness for symptomatic infection and hospital admission among children aged 6–11 vaccinated with CoronaVac.** The dots represent the adjusted vaccine effectiveness (VE;1- adjusted odds ratio) estimates (sample $n = 197,958$), with error bars indicating the corresponding 95% Wald's C.I. for symptomatic infection and Profile's likelihood C.I. for hospital admission. Red represents adjusted VE against symptomatic infection, and blue against hospital admission considering vaccination status (in days post first and second dose). The comparison group was the unvaccinated.

A more detailed description from our database can be found in the Supplementary Materials. In addition, we deterministically linked the data using the information provided by DATASUS from the Brazilian Ministry of Health. Dataset quality assessment and linkage details have been described before[18–21].

## Study design

We used a test-negative design, which is a type of case-control study among the population tested, with controls selected from those who presented a negative test[22]. The study population comprised childrens aged 6–11 years with COVID-19-related symptoms in Brazil from January 21, 2022, to April 15, 2022, with a predominant circulation of the Omicron variant (>98% of sequenced viruses)[23]. We linked records of SARS-CoV-2 reverse transcription-polymerase chain reaction (RT-PCR) and antigen tests to national vaccination and clinical records. Participants were symptomatic children with a sample collected within ten days of symptom onset. Cases of confirmed infection were those with a positive SARS-CoV-2 RT-PCR or antigen test, and control had a negative SARS-CoV-2 RT-PCR or antigen test. Additionally, we evaluated severe COVID-19 (hospital admission), defined as a positive test that occurred within 14 days before the hospitalisation date and up to four days after hospital admission, and death occurring within 28 days after a positive test.

We excluded: (1) individuals older than 11 years and younger than 6 years; (2) individuals who received vaccines other than CoronaVac; and (3) tests among asymptomatic people and tests referring to a symptom onset date after the notification date; (4) individuals whose time interval between the first and second doses was less than 14 days and received first dose before January 21, 2022; (5) negative test within 14 days of a previous negative test; (6) negative test followed by a positive test up to 7 days; (7) any test after a positive test up to 90 days, and (8) tests with missing information on age, sex, city of residence, sample collection, or first symptoms date; (9) any individual which received the third dose. Our exposure was vaccination status stratified by the time since the last dose on the date of sample collection, categorised as: unvaccinated and, for the vaccinated, grouped in periods (days) after each dose: first dose (0–13 days, and ≥14 days), second dose (0–13, ≥14 days). In addition, the following confounders were included in the model: age, gender, ethnicity, time (month), region of residence, socioeconomic position measured by quintile of deprivation (the Índice Brasileiro de Privação in Brazil)[24], previous SARS-CoV-2 infection (between 3–6 months or more six months ago), number of

comorbidities commonly associated with COVID-19 illness. The odds ratio (OR) comparing the odds of vaccination between cases and controls and its associated 95% Confidence Interval (CI) were derived using logistic regression. VE was estimated as (1-OR)*100, obtained from an adjusted model including the described covariates, expressed as a percentage. All data processing and analyses were performed in R (version 4.1.1)[25], using the Tidyverse package[26]. Missing values relating to ethnicity were imputed using multiple imputations, as sensitivity analyses. For these analyses, we used the MICE package (version 1.16) with five imputations[27]. We conducted a logistic regression for rare events (ICU admission) using Firth's bias reduction method (Logistf package v. 1.24.1)[28].

We followed the RECORD reporting guidelines (Table S5)[29]. The statistical analysis plan (SAP) was published in https://vigivac.fiocruz.br/. The Brazilian National Commission in Research Ethics approved the research protocol (CONEP approval number 4.921.308) and (CAAE registration no. 50199321.9.0000.0040). CONEP waived the requirement for informed consent because we did not have access to identified data. The Brazilian Ministry of Health authorized the use of these data by the Vaccination Digital Vigilance (VigiVac) program under the data protection law which allows such a consent for public health research.

## Data availability

Our statistical analysis plan is available at https://vigivac.fiocruz.br. Regarding Brazilian data availability, one of the study coordinators (M.B.-N.) signed a term of responsibility on using each database made available by the Ministry of Health (MoH). Each member of the research team signed a term of confidentiality before accessing the data. Data was manipulated in a secure computing environment, ensuring protection against data leakage. The Brazilian National Commission in Research Ethics approved the research protocol (CONEP approval no. 4.921.308). Our agreement with the MoH for accessing the databases patently denies authorization of access to a third party. Any information for assessing the databases must be addressed to the Brazilian MoH at https://datasus.saude.gov.br/, and requests can be addressed to datasus@saude.gov.br. In this study, we used anonymized secondary data following the Brazilian Personal Data Protection General Law, but it is vulnerable to re-identification by third parties as they contain dates of relevant health events regarding the same person. To protect the research participants' privacy, the approved Research Protocol (CONEP approval no. 4.921.308)

authorises the dissemination only of aggregated data, such as the data presented here.

## Code availability

All code used in this study is publicly available at https://github.com/cidacslab/vigivac/tree/main/tnd_02.

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

## Acknowledgements

This study is part of the Fiocruz VigiVac programme and the authors acknowledge DATASUS for its diligent work in providing the unidentified Brazilian databases. This study was partially supported by a donation from the "Fazer o bem faz bem" program, from JBS, S.A, GLW, MLB, VB and M.B-N. are research fellows from the Brazilian National Research Council. M.B-N. acknowledges Fundação de Amparo à Pesquisa do Estado da Bahia (FAPESB)–Grant PNX0008/2014/ Fapesb, Edital 08/2014–Programa de Apoio a Núcleos de Excelência. G.L.W. acknowledges Fundação Carlos Chagas Filho de Amparo à Pesquisa do Estado do Rio de Janeiro (E-26/210.180/2020). ESP is funded by the Wellcome Trust [Grant number 213589/Z/18/Z]. E.P.S. is funded by the Wellcome Trust [Grant number 213589/Z/18/Z]. The funders had no role in study design, data collection, data analysis, data interpretation, the report's writing, or decision to publish. For the purpose of Open Access, the author has applied a CC BY public copyright licence to any Author Accepted Manuscript (AAM) version arising from this submission.

## Author contributions

E.S.P., M.L.B and M.B.-N. conceived the idea for the study. All authors contributed to the study design, with P.T.V.F., E.S.P., A.G.J and T.C.-S. drafting the statistical analysis plan. P.T.V.F. conducted the statistical analysis, E.S.P. checked the analysis code. P.T.V.F., J.B.S.J., T.C.-S. and V.dAO. had access to individual-level data for Brazil and performed data linkage. M.B.-N., V.dAO. and M.L.B. organised the data linkage and secured funding. E.S.P., P.T.V.F and F.J.O.A. wrote the initial draft of the manuscript. E.S.P., L.R., G.L.W., G.O.P., M.L.B., V.B., N.P., and M.B.-N. critically revised the manuscript. PTVF and VdAO accessed and verified the data and analyses. All authors critically revised the manuscript and approved the final version for submission.

## Competing interests

M.B.-N. reports grants from the Fazer o bem faz bem program from JBS. S.A. V.dA.O., V.B., M.L.B., and M.B.-N. are employees of Fiocruz, a federal public institution, which manufactures Vaxzevria in Brazil, through a full technology transfer agreement with AstraZeneca. Fiocruz allocates all its manufactured products to the Ministry of Health for public health use. The remaining authors declare no competing interests.
