## [Peer Review File · Nature Communications]

Vaccine effectiveness of CoronaVac against COVID-19 among children in Brazil during the Omicron periodREVIEWER COMMENTS

Reviewer #1 (Remarks to the Author):

General comments

The paper provides vaccine effectiveness (VE) estimates for CoronaVac in children between 6 to 11 years, based on information gathered between January 21 and April 19, 2022, where a predominant circulation of the Omicron variant was observed (>50% of sequenced viruses). The authors conducted a test-negative design to assess VE against infection and severe outcomes. The severe outcome is defined as hospitalisation or death.

The relevance of these findings cannot be understated given the mass production of this vaccine and its availability in LMIC globally. This information can inform policy makers and provide supportive evidence to the public at large. The majority of the literature on VE is about mRNA vaccines. This manuscript provides an important contribution and shows that the CoronaVac provides protection for children against SARS-CoV-2 infection and severer outcomes.

The paper is in general well written and easy to follow. However, I would advice the authors to carefully proofread the manuscript as there are still typos in its current version (e.g. lines 39-40: "For severe outcomes (hospitalisation or death) VE was 69.2% (95% CI: 11.7-93.6) at 0-13 days and 63.5% (95% 40 CI: 5.8-90.0).")

Specific comments:

- 1.- Lines 84 - 87: I don't think that Jara, A. et al. (2021. Effectiveness of an Inactivated SARS-CoV-2 Vaccine in Chile. New England Journal of Medicine 385: 875-884) is the correct reference for that sentence. Please correct if this is needed.
- 2.- The authors stated that a limitation of the study "is the under ascertainment of the previous infection once there is a high rate of asymptomatic infection in children, which may not be notified." Please discuss the potential implication of this statement on the provided VE estimates.
- 3.- The authors considered hospitalisation or death as a single outcome. Could you please provide VE estimates for each of them separately. Also, why not to consider ICU admission as an outcome of interest?
- 4.- The authors perform a complete case analysis by excluding tests with missing information about age, sex, city of residence, sample collection, or first symptoms date. The authors should provide information on the missing data structure, justify whether the missing at random assumption is valid (e.g. for city of residence), and also provide VE estimates based on multiple imputation.
- 5.- If I understood correctly, the authors excluded from the analyses to children that had received a third dose, which could induce bias in the VE estimates. Could you please confirm that the information of children infected between the second and third dose was not employed in the analysis?
- 6.- The authors evaluated severe COVID-19 (hospital admission or death), defined as a positive test that occurred within 14 days before the hospitalisation date and up to three days after hospital admission, and death occurring within 28 days after a positive test. Could you please clarify how information of children having positive tests > 14 days before the hospitalisation date was considered in the study? Could you please also clarify how deaths occurring after 28 days after a positive test were considered in the study?

7.- The authors considered previous SARS-CoV-2 infection experience as an additional correction factor in the analyses. I do not think this is the correct way to include this information as VE heterogeneity can be expected in children with or without previous SARS-CoV-2 infection experience. Please provide separate VE estimates for children with and without previous without previous SARS-CoV-2 infection experience.

8.- Related to the previous point, it is not clear to me how children with previous infection between 0 - 2 and more than six months ago were considered in the analyses.

9. Please provide information on the distribution of the number of days since completion of the primary vaccination scheme. If the VE estimates are based on information shortly after completion of the primary vaccination scheme, they could be very much the optimal VE to be observed, which will wane rapidly over the coming months, as we know now for Omicron. If this is the case, this point needs to be emphasized in the manuscript.

Reviewer #2 (Remarks to the Author):

This study presents data on the effectiveness of the CoronaVac vaccine in children against omicron. With such a paucity of information on vaccine effectiveness in this age group, the findings address an important data gap; doing so could provide helpful evidence for any public health programmes considering whether to provide vaccination for children.

I have the following comments, mainly about clarification:

While the dataset is impressive, it would be very helpful to have more information about the testing practices underlying the data. For example, what proportion of symptomatics were tested in healthcare settings, as opposed to community-level testing or contact tracing?

Depending on the answer to the above question:

- If the great majority of symptomatics were tested in healthcare settings, I would recommend qualifying all references to VE against 'symptomatic' illness as being against 'medically attended symptomatic illness' instead.
- If instead there is a balanced mix of settings, it would be helpful to include a sub-analysis concentrating only on medically attended illness. With one of the key benefits of vaccination being to reduce burden on healthcare services, this sub-analysis would speak directly to that aim.

It would also be good to have a couple of lines as background on the database: for example, what proportion of the population does it cover (likely quite high)? What is it that makes this database 'high quality' for the purpose of the current analysis?

Minor comments

Line 86: Suggest replacing 'population' by 'demographic' to avoid confusion.

Line 93: Please define TND and include further details in the Methods or Study Design.

Line 104: To avoid a point estimate being misleading, I suggest rewording as "...against severe disease, albeit with wide uncertainty intervals."

Line 123: Could you specify exactly what proportion was Omicron? Over 50% does not seem like a large majority.

REVIEWER COMMENTS

Reviewer #1 (Remarks to the Author):

General comments

The paper provides vaccine effectiveness (VE) estimates for CoronaVac in children between 6 to 11 years, based on information gathered between January 21 and April 19, 2022, where a predominant circulation of the Omicron variant was observed (>50% of sequenced viruses). The authors conducted a test-negative design to assess VE against infection and severe outcomes. The severe outcome is defined as hospitalisation or death.

The relevance of these findings cannot be understated given the mass production of this vaccine and its availability in LMIC globally. This information can inform policy makers and provide supportive evidence to the public at large. The majority of the literature on VE is about mRNA vaccines. This manuscript provides an important contribution and shows that the CoronaVac provides protection for children against SARS-CoV-2 infection and severer outcomes.

The paper is in general well written and easy to follow. However, I would advice the authors to carefully proofread the manuscript as there are still typos in its current version (e.g. lines 39-40: "For severe outcomes (hospitalisation or death) VE was 69.2% (95% CI: 11.7-93.6) at 0-13 days and 63.5% (95% 40 CI: 5.8-90.0).")

A: We appreciate the reviewer comments on our manuscript. We have proofread the manuscript.

Specific comments:

1.- Lines 84 - 87: I don't think that Jara, A. et al. (2021. Effectiveness of an Inactivated SARS-CoV-2 Vaccine in Chile. New England Journal of Medicine 385: 875–884) is the correct reference for that sentence. Please correct if this is needed.

A: Thank you for noticing. We have modified the reference to the correct reference below. (Line: 88; Page: 3).

Reference #6: Jara A, Undurraga EA, Flores JC, et al. Effectiveness of an Inactivated SARS-CoV-2 Vaccine in Children and Adolescents: A Large-Scale Observational Study. SSRN Preprint:<http://dx.doi.org/10.2139/ssrn.4035405>.

2.- The authors stated that a limitation of the study "is the under ascertainment of the previous infection once there is a high rate of asymptomatic infection in children, which may not be notified." Please discuss the potential implication of this statement on the provided VE estimates.

A: The high rates of asymptomatic infections could lead to misclassifying this variable. However, these errors probably affected both vaccinated and unvaccinated children and, therefore, unlikely to bias the results of this study. Suppose this assumption is incorrect and previous infection was underestimated differently by vaccination status. In that case, it could bias the results, depending on whether it was underestimated in the vaccinated group or underestimated in the unvaccinated group. We have added this point is now included to the limitation paragraph (Line: 102-106 Page: 4).

“Due to high rates of asymptomatic infection allied to limited testing in Brazil among children since the database from the study only accounts for register from the healthcare system and not community testing. The under ascertainment of previous infection may bias the VE estimates if this condition occurs differentially in the vaccinated and unvaccinated group”

3.- The authors considered hospitalisation or death as a single outcome. Could you please provide VE estimates for each of them separately. Also, why not to consider ICU admission as an outcome of interest?

A: We thank the reviewer comment. Indeed, our initial plan of analyses was to analyse them separately. Unfortunately for the study, the number of ICU admissions and deaths among children with two vaccine doses was too small to perform the separated analyses.

4.- The authors perform a complete case analysis by excluding tests with missing information about age, sex, city of residence, sample collection, or first symptoms date. The authors should provide information on the missing data structure, justify whether the

missing at random assumption is valid (e.g. for city of residence), and also provide VE estimates based on multiple imputation.

A: In Brazil, ethnicity has been historically associated with social-economic conditions, and therefore we adjusted our model using this variable as a covariate. However, the ethnicity register is optional, and we have about 20% of missing data. To overcome this issue and still include this important covariate, we have treated the missing data as a category within those existing. We understand the referee point and we have now performed a sensitivity analysis using multiple imputations for ethnicity variable. The results were similar to the primary analyses and are included in the supplementary material (Table S2), we have also added a description to the method section (Line: 161-163; Page: 6). The other variable (Deprivation Index and city of residence) had a small percentage of missing values (<0.1%) which were excluded to perform the regression model.

5.- If I understood correctly, the authors excluded from the analyses to children that had received a third dose, which could induce bias in the VE estimates. Could you please confirm that the information of children infected between the second and third dose was not employed in the analysis?

A: Indeed, we have not employed children infected between the second and third dose in our analysis, since we exclude all the children that received a third dose. The number of children with booster dose was 438, which represent >0.1% of the total study group, and this is unlikely to influence results. This exclusion criterion can be found in the Methods section (Line: 150; Page: 5).

6.- The authors evaluated severe COVID-19 (hospital admission or death), defined as a positive test that occurred within 14 days before the hospitalisation date and up to three days after hospital admission, and death occurring within 28 days after a positive test. Could you please clarify how information of children having positive tests > 14 days before the hospitalisation date was considered in the study? Could you please also clarify how deaths occurring after 28 days after a positive test were considered in the study?

A: We understand the reviewer concern. The criterion that positive test within 14 days before hospitalisation and death less than 28 days from a positive test, was applied to

ensure that for each individual test, hospital admission and death were caused by the same SARS-CoV-2 episode.

7.- The authors considered previous SARS-CoV-2 infection experience as an additional correction factor in the analyses. I do not think this is the correct way to include this information as VE heterogeneity can be expected in children with or without previous SARS-CoV-2 infection experience. Please provide separate VE estimates for children with and without previous without previous SARS-CoV-2 infection experience.

A: We understand the reviewer's concern. We performed the analyses stratified among those without notified previous infection, and the results were similar to the primary analyses. Due to small numbers, we could not perform the analysis among those with a previous infection.

Without Previous Infected		VE% (95% CI)			
Vaccination Status	Negative tests	Positive tests Infection	Positive tests Severe	Symptomatic Infection	Severe Outcomes
Unvaccinated	68,226	71,703 (51.24%)	434 (0.63%)		
1st dose					
0-13 days	6,699	7,419 (52.55%)	32 (0.48%)	[-97 (-13.9, -5.6)]	22.4 (-10.7,47.4)
≥14-2nd dose	19,657	8,136 (29.27%)	44 (0.22%)	20.4 (17.7,23.0)	44.1 (23.0,60.4)
2nd dose					
0-13 days	4,402	620 (12.35%)	2 (0.05%)	30.3 (23.6,36.5)	82.1 (43.0,97.1)
≥14 days	6,608	508 (7.14%)	6 (0.09%)	40.3 (34.2,45.9)	58.0 (8.6, 84.0)

8.- Related to the previous point, it is not clear to me how children with previous infection between 0 - 2 and more than six months ago were considered in the analyses.

A: In our model we adjust OR with previous infections registers occurring between 3-6 months and more than 6 months prior to any test (positive or negative) (Line: 156; Page: 6). The number of tests included with previous infection are in Table S2. Any infection occurring between 0-3 months are excluded from our analysis in case of a test positive as they may represent the same SARS-CoV-2-episode.

9. Please provide information on the distribution of the number of days since completion of the primary vaccination scheme. If the VE estimates are based on information shortly after completion of the primary vaccination scheme, they could be very much the optimal VE to be observed, which will wane rapidly over the coming months, as we know now

for Omicron. If this is the case, this point needs to be emphasized in the manuscript.

A: You are right, indeed the VE estimated in this study is probably the optimal VE which will wane over time. Since our follow-up time post-second dose is still short, we have now included this point in the discussion (Line: 92-96; Page: 3-4). Also, we have provided a chart with the distribution of time between second dose and collection date to the supplementary materials (Figure S2).

Reviewer #2 (Remarks to the Author):

This study presents data on the effectiveness of the CoronaVac vaccine in children against omicron. With such a paucity of information on vaccine effectiveness in this age group, the findings address an important data gap; doing so could provide helpful evidence for any public health programmes considering whether to provide vaccination for children.

A: We thank the reviewer comments on our paper.

I have the following comments, mainly about clarification:

While the dataset is impressive, it would be very helpful to have more information about the testing practices underlying the data. For example, what proportion of symptomatics were tested in healthcare settings, as opposed to community-level testing, or contact tracing?

A: We understand the point made by the reviewer. In Brazil, our dataset presents only the registers of individuals which were tested in the public or private healthcare facilities. Since this information was not clear in the text, we have added a brief explanation to methods section (Line:120-122; Page: 4-5).

Depending on the answer to the above question:

- If the great majority of symptomatics were tested in healthcare settings, I would recommend qualifying all references to VE against 'symptomatic' illness as being against 'medically attended symptomatic illness' instead.

A: We agree and have changed the text accordingly.

- If instead there is a balanced mix of settings, it would be helpful to include a sub-analysis concentrating only on medically attended illness. With one of the key benefits of vaccination being to reduce burden on healthcare services, this sub-analysis would speak directly to that aim.

- It would also be good to have a couple of lines as background on the database: for example, what proportion of the population does it cover (likely quite high)? What is it that makes this database 'high quality' for the purpose of the current analysis?

A: The reviewer raises an important point. Since the beginning of the pandemics the Brazilian Health Ministry determined that all notifications of suspected COVID-19 cases and severe acute respiratory syndrome must be reported through two different systems: e-SUS Notifica and SIVEP-Gripe. Therefore, all Brazilians that attended the private or public healthcare system are notified in those system.

The databases used in the analyses are supplied by the Brazilian Health Ministry and assemble all registers of individuals notified with specific related-COVID-19 symptoms. The two databases used in our study are:

1) The e-SUS Notifica was created in March 2020 with the goal to follow the suspected cases of COVID-19 in Brazil. In this database, there are only registers of individuals with mild/moderate COVID-related symptoms.

2) The SIVEP-Gripe was created after the influenza pandemic of 2009 and expanded to include COVID-19. The case definition in this database is individuals with severe acute respiratory syndrome (SARS) who present dyspnoea/respiratory discomfort, persistent pressure or pain in the chest and oxygen saturation less than 95% without oxygen or cyanosis of the lips or face.

Also, individuals who died from SARS illness independent of hospitalisation are registered in SIVEP-Gripe or e-SUS-Notifica. In these databases we have the information of SARS-CoV-2 testing and the individual's related symptoms.

To assemble our dataset, we link the notifications of these two databases, with the vaccination registers from National COVID-19 vaccination campaign using an unique anonymised ID by individual. Our database was used in other publications from our group.

We have noticed that this information was clear in our manuscript, and we have now added a more detailed explanation of our database (Supplementary materials) and references to studies that used the same database to analyse VE in Brazil.

References:

13. Katikireddi, S. V. *et al.* Two-dose ChAdOx1 nCoV-19 vaccine protection against COVID-19 hospital admissions and deaths over time : a retrospective , population-based cohort study in Scotland and Brazil. *The Lancet* **399**, 25–35 (2022).
14. Cerqueira-Silva, T. *et al.* Vaccine effectiveness of heterologous CoronaVac plus BNT162b2 in Brazil. *Nature Medicine* **28**, (2022).
15. Cerqueira-Silva, T. *et al.* Vaccination plus previous infection: protection during the omicron wave in Brazil. *The Lancet Infectious Diseases* **Correspond**, 1–2 (2022).
16. Cerqueira-Silva, T. *et al.* Effectiveness of CoronaVac, ChAdOx1 nCoV-19, BNT162b2, and Ad26.COV2.S among individuals with previous SARS-CoV-2 infection in Brazil: a test-negative, case-control study. *The Lancet Infectious Diseases* **22**, 791–801 (2022).

Minor comments

Line 86: Suggest replacing ‘population’ by ‘demographic’ to avoid confusion.

A: We agree and have changed the text accordingly (Line: 87; Page: 3).

Line 93: Please define TND and include further details in the Methods or Study Design.

A: Thank you. The definition of TND and further details are now included in the Methods section (Line:131-132; Page: 4).

Line 104: To avoid a point estimate being misleading, I suggest rewording as “...against severe disease, albeit with wide uncertainty intervals.”

A: Thank you. We have now modified the text accordingly (Line: 111; Page: 4).

Line 123: Could you specify exactly what proportion was Omicron? Over 50% does not seem like a large majority.

A: Thank you for noticing. Our study period started on January 21, 2022, at this period the Omicron variant already represented $\geq 98.7\%$ of the sequenced samples in Brazil. (Ref: <http://www.genomahcov.fiocruz.br/dashboard-en/>). We have now revised this information in the text (Line: 134; Page: 5).

REVIEWERS' COMMENTS

Reviewer #1 (Remarks to the Author):

The authors have done a good job in replying to some of my previous concerns. However, I am not fully satisfied with some of the responses to my previous points. Therefore, I would ask the authors to take another look at the following points:

1.- In response to my comment on the potential effect of the under ascertainment of the previous infection on the VE estimates, the authors included the following discussion in the revised version of the manuscript: "Due to high rates of asymptomatic infection allied to limited testing in Brazil among children since the database from the study only accounts for register from the healthcare system and not community testing. The under ascertainment of previous infection may bias the VE estimates if this condition occurs differentially in the vaccinated and unvaccinated group". I disagree with this conclusion as VE estimators could be bias regardless the differential under ascertainment rate in both groups. Please see, e.g., Lipsitch et al., (2019. Depletion-of-susceptibles bias in influenza vaccine waning studies: how to ensure robust results, *Epidemiology and Infection*, 147: e306), for a discussion on this important aspect. The bigger problem here can be the non-detected reduction of susceptibles cases in the unvaccinated group, which can generates an important underestimation of the VE.

2.- The authors considered hospitalisation or death as a single outcome. In their response to my comment on this point they mention that: "Unfortunately for the study, the number of ICU admissions and deaths among children with two vaccine doses was too small to perform the separated analyses." This should be emphasized in the manuscript. Also, it is not clear to me why an exact logistic regression analysis cannot be performed for this data. I do think that is really important to provide VE estimates for each outcome. The exact logistic regression analysis could also help to reduce the high observed asymptotic CIs presented by the authors for the severe outcomes.

3.- The authors reply that "due to small numbers, we could not perform the analysis among those with a previous infection." If this is the case, I do not understand why the authors do not present the analysis based on those without notified previous infection only. Again, an exact logistic regression could also be employed for this analysis.

REVIEWERS' COMMENTS

Reviewer #1 (Remarks to the Author):

The authors have done a good job in replying to some of my previous concerns. However, I am not fully satisfied with some of the responses to my previous points. Therefore, I would ask the authors to take another look at the following points:

A: We appreciate the reviewer's comments.

1.- In response to my comment on the potential effect of the under ascertainment of the previous infection on the VE estimates, the authors included the following discussion in the revised version of the manuscript: "Due to high rates of asymptomatic infection allied to limited testing in Brazil among children since the database from the study only accounts for register from the healthcare system and not community testing. The under ascertainment of previous infection may bias the VE estimates if this condition occurs differentially in the vaccinated and unvaccinated group". I disagree with this conclusion as VE estimators could be bias regardless the differential under ascertainment rate in both groups. Please see, e.g., Lipsitch et al., (2019. Depletion-of-susceptibles bias in influenza vaccine waning studies: how to ensure robust results, *Epidemiology and Infection*, 147: e306), for a discussion on this important aspect. The bigger problem here can be the non-detected reduction of susceptibles cases in the unvaccinated group, which can generates an important underestimation of the VE.

A: We agree with the reviewer. We have now added to the limitation paragraph in the Discussion section the following text (Line: 107-109; Page: 4):

"Also, the under ascertainment of previous infection may bias the VE estimates if this condition occurs differentially or non-differentially in the vaccinated and unvaccinated group (26)."

26 - Lipsitch, M., Goldstein, E., Ray, G. T. & Fireman, B. Depletion-of-susceptibles bias in influenza vaccine waning studies: how to ensure robust results. *Epidemiology and infection* **147**, e306 (2019).

2.- The authors considered hospitalisation or death as a single outcome. In their response to my comment on this point they mention that: "Unfortunately for the study, the number of ICU admissions and deaths among children with two vaccine doses was too small to perform the separated analyses." This should be emphasized in the manuscript. Also, it is not clear to me why an exact logistic regression analysis cannot be performed for this data. I do think that is really important to provide VE estimates for each outcome. The exact logistic regression analysis could also help to reduce the high observed asymptotic CIs presented by the authors for the severe outcomes.

A: We have now separated the three main outcomes for severe diseases: Hospital admission, ICU admission and Death outcome. To analyse the outcomes for rare events

we have conducted a Firth bias methods (<https://www.sas.com/content/dam/SAS/support/en/sas-global-forum-proceedings/2020/4654-2020.pdf>, <https://www3.nd.edu/~rwilliam/stats3/rareevents.pdf>) (1) using Logistf package in R <https://cran.r-project.org/web/packages/logistf/logistf.pdf>. The table below shows the number of cases for each of the outcomes separately. As we can see we don't have any death case after the second dose, and for ICU admission we only have 2 cases 14 days post-second dose. Considering these results, we have now modified our main table (Table 1) including only hospital admission events instead of including death events together. We have also clarified in the text that we don't have death events after the second dose. Furthermore, we have provided a supplementary table (Table S2) including the estimates of ICU admission event separately using the Firth Bias method.

Vaccination Status	Hospital admission n = 508	ICU admission n = 108	Death n = 39	negative tests n = 108,363
Unvaccinated	428 (0.61%)	88 (0.13%)	35 (0.05%)	69,923
1st dose				
0-13 days	30 (0.44%)	7 (0.10%)	2 (0.03%)	6,862
≥14-2nd dose	42 (0.21%)	10 (0.05%)	2 (0.01%)	20,193
2nd dose				
0-13 days	2 (0.04%)	1 (0.02%)	0 (0.00%)	4,552
≥14 days	6 (0.09%)	2 (0.03%)	0 (0.00%)	6,833

(1) Firth, D. "Bias reduction of maximum likelihood estimates." *Biometrika* **82**, 667 (1995).

3.- The authors reply that "due to small numbers, we could not perform the analysis among those with a previous infection." If this is the case, I do not understand why the authors do not present the analysis based on those without notified previous infection only. Again, an exact logistic regression could also be employed for this analysis.

A: We understand the reviewer's concern. We have now added to the supplementary materials the analysis excluding the previous infected individuals (Table S4). The results were similar to the main analyses; therefore, we kept the primary analyses as principal and the analyses excluding the previously infected as supplementary material.